# Effectiveness and Safety of Palbociclib Plus Endocrine Therapy in Patients with Advanced Breast Cancer: A Multi-Center Study in China

**DOI:** 10.3390/cancers15174360

**Published:** 2023-09-01

**Authors:** Xinyu Wu, Nan Jin, Hongfei Gao, Min Yan, Qianjun Chen, Tao Sun, Chunfang Hao, Yanxia Zhao, Xinhua Han, Yueyin Pan, Xiang Huang, Wei Li, Kun Wang, Yongmei Yin

**Affiliations:** 1Department of Medicine, Nanjing Medical University, Nanjing 210029, China; wuxy@njmu.edu.cn (X.W.); queenie.jn@outlook.com (N.J.); lorelai@njmu.edu.cn (X.H.); real.lw@163.com (W.L.); 2Department of Oncology, The First Affiliated Hospital of Nanjing Medical University, Nanjing 210029, China; 3Department of Breast Cancer, Cancer Center, Guangdong Provincial People’s Hospital, Southern Medical University, Guangzhou 510515, China; nfzj1988@163.com; 4Department of Breast Disease, Henan Breast Cancer Center, The Affiliated Cancer Hospital of Zhengzhou University, Zhengzhou 450008, China; ym200678@126.com; 5Department of Breast Disease, Guangdong Provincial Hospital of Chinese Medicine, Guangzhou 510120, China; cqj55@163.com; 6Department of Medical Oncology, Liaoning Cancer Hospital & Institute, Shenyang 110042, China; jianong@126.com; 7Department of Breast Oncology, Tianjin Medical University Cancer Institute and Hospital, Tianjin 300060, China; haochunfang@tjmuch.com; 8Union Hospital, Tongji Medical College, Huazhong University of Science and Technology, Wuhan 430074, China; sophia7781@126.com; 9Division of Life Science and Medicine, Department of Oncology, The First Affiliated Hospital of University of Science and Technology of China (USTC), University of Science and Technology of China, Hefei 230026, China; lnn279@ustc.edu.cn (X.H.); panyueyin2021@163.com (Y.P.)

**Keywords:** palbociclib, metastatic breast cancer, real-world study

## Abstract

**Simple Summary:**

Palbociclib is one of the preferred treatments for hormone-receptor-positive, HER2-negative metastatic breast cancer in women; its effectiveness and safety in the Chinese real-world population require further investigation. Our study aimed to identify the clinical outcomes of patients who received palbociclib in combination with endocrine therapy (ET) in China. The effectiveness of palbociclib plus ET in treating metastatic breast cancer was confirmed in 397 Chinese patients across eight clinical sites. The regimens were well tolerated by both the general and elderly groups. However, some factors were identified as potential hindrances to the therapy’s benefits, including higher Ki-67 expression, primary resistance to ETs, liver metastases, more metastatic sites, later line of therapy, and the use of fulvestrant instead of aromatase inhibitors. Palbociclib is useful for treatment of metastasis breast cancer, and our study may provide a basis for further research.

**Abstract:**

Background: Palbociclib has been approved for marketing in China. However, its effectiveness, safety, and latent variables in the Chinese population require further investigation. Methods: Information was retrieved from 397 patients with metastatic breast cancer (mBC) who received at least two cycles of palbociclib plus endocrine therapy (PAL plus ET) at eight clinical sites in China. The patients’ demographic characteristics, treatment patterns, and adverse events (AEs) were analyzed. Results: The objective response rate (ORR) and clinical benefit rate (CBR) for PAL plus ET were 28.97% and 66.25%, respectively. The median PFS was 14.2 months in the whole population. In addition to protein Ki-67 status and sensitivity to ETs, no liver metastases, fewer metastatic sites, an earlier line of therapy, and treatment combined with AI instead of FUL were also considered as independent prognostic factors for PAL treatment. Administration of PAL was generally well tolerated in patients with hormone-receptor-positive and human-epidermal-growth-factor-receptor-2-negative (HR+/HER2−) advanced breast cancer (ABC). The therapy was safe in the elderly population, which is consistent with the outcomes of the whole population and previous reports. Conclusions: In this most widely distributed study in China to date, palbociclib combined with ET proved its effectiveness for HR+/HER2− ABC treatment, and adverse events were manageable. Here, we identified some independent prognosis factors, but the mechanism by which these factors influence effectiveness requires further verification.

## 1. Introduction

Hormone receptor (HR)-positive and human epidermal growth factor receptor-2 (HER2)-negative (HR+/HER2−) breast cancer is one of the most commonly diagnosed cancers in women and generally has a favorable prognosis compared to other subtypes of breast cancer [1]. Although many patients with this type of breast cancer do not develop metastatic disease, some cases either harbor distant metastasis at the time of diagnosis or develop metastatic lesions over time [1].

Cyclin-dependent kinase 4/6 (CDK 4/6) inhibitors alter the implications of HR+/HER2− ABC. The CDK 4/6 pathway primarily regulates the progression of the cell cycle from the G1 (pre-DNA synthesis) to S (DNA synthesis) phase. CDK 4/6 inhibitors (CDK 4/6i) target the CDK enzyme complex, disrupt the cell cycle, and prevent uncontrolled cancer [2]. Multiple clinical trials found statistically significant improvements in progression-free survival (PFS) and overall survival (OS) when CDK 4/6 inhibitors were added to endocrine therapy. Palbociclib (PAL) is an oral CDK 4/6 inhibitor, and several well-designed randomized controlled trials have evaluated its efficacy in the treatment of HR+/HER2− mBC. The PALOMA-2 clinical trial showed that adding PAL to letrozole (LET) significantly improved PFS from 14.5 months to 24.8 months (*p* < 0.001) [3,4]. The PALOMA-3 trial proved that the combination of PAL and fulvestrant (FUL) improved PFS compared to placebo plus FUL (9.5 months vs. 4.6 months, *p* < 0.0001) [5,6]. PALOMA-4, a clinical trial involving the Asian population which considered ethnic difference, also supported this conclusion, as the median PFS was 21.5 months in the PAL–LET group compared to 13.9 months in the placebo–LET group (*p* = 0.0012) [7]. Based on these results, the National Comprehensive Cancer Network (NCC) and Chinese Society of Clinical Oncology (CSCO) recommend CDK 4/6i as a viable treatment option in combination with endocrine therapy (ET) [8,9].

Palbociclib has been marketed in China for more than five years. Some studies have investigated the effectiveness and safety of palbociclib in China, yet they were limited by enrollment and regional distributions, and the results were not always consistent [10,11,12]. Unlike well-controlled clinical trials, the real-world population is much more heterogeneous and requires a large population to represent clinical practice with PAL treatment. Data on drug use, related outcome indicators, and availability of patient database resources in various hospitals make database-based clinical research possible. Therefore, this study aimed to elucidate the application of PAL in China, describe its clinical outcomes and safety, and investigate its factors in the Chinese population. In addition, this study aimed to validate the effectiveness and potential toxicity of CDK 4/6i in the geriatric population or patients with chronic disease, expand the information on Chinese patients, and provide more references for the treatment of mBC.

## 2. Materials and Methods

### 2.1. Study Design

This study was designed as a non-interventional, retrospective, multi-center clinical study that involved patients with mBC who received at least two cycles of palbociclib in eight clinical sites in China from July 2016 to October 2022 (Appendix A). The sample size was calculated using the 10 events per variable (EPV) suggestion; the minimum sample size estimate was 378 patients [13]. The study was performed according to the ethical standards of the responsible institutional committee on human experimentation, the principles of the Declaration of Helsinki, and all applicable national regulations. Informed consent was obtained from all the enrolled patients.

### 2.2. Patients

A total of 397 patients who met the following criteria between July 2016 and October 2022 from eight public oncology clinics were included in this retrospective study: (1) patients who had a histopathologically or imaging-confirmed diagnosis of invasive locally advanced or metastatic breast cancer; (2) patients with pathologically confirmed hormone-receptor-positive and HER2-negative mBC; (3) patients who had at least three well-documented medical records at the participating clinical center, including at least one hospitalization record; (4) patients who had at least one extracranial measurable lesion or osteolytic or mixed bone metastases based on the Response Evaluation Criteria in Solid Tumors v. 1.1 (RECIST 1.1) [14]; (5) patients with an Eastern Cooperative Oncology Group (ECOG) performance status [15] of 3 or less. The exclusion criteria were as follows: (1) patients who were <18 years old; (2) patients with other concurrent cancers; (3) male breast cancer; (4) patients with bilateral breast cancer; (5) patients who received palbociclib as a neoadjuvant regimen; and (6) patients with unknown or ambiguous information.

The eighth TNM staging system by the American Joint Committee on Cancer (AJCC) was used to determine the clinical stage [16]. Hormone receptor (HR) positivity was defined as positivity for estrogen receptor (ER) and/or progesterone receptor (PR) [17]. The expression of human epidermal growth factor receptor-2 (HER2) was determined based on the guidelines of the American Society of Clinical Oncology/College of American Pathologists. HER2 was considered positive using 3+ immunohistochemical staining or amplified fluorescence in situ hybridization (FISH) [17,18]. ER and PR positivity were confirmed by immunohistochemistry (IHC); staining of more than 1% of tumor cells was considered positive [17,19]. The study population was divided into two groups based on these evaluations. Luminal A subtype was defined as ER positive with high expression of PR (more than 20% of tumor cells staining positive) and HER2 negative. The luminal B subtype was determined as ER positive with low expression of PR (positive staining in <20% of tumor cells) and HER2 negative. A Ki-67 index of less than 15%, 15% to less than 30%, and ≥30% was defined as low, medium, or high Ki-67 expression, respectively [9]. Clinicopathological characteristics and treatment data were collected from patients’ medical records. The definition of sensitivity to ET was consistent with ABC6 [20]. ET-naïve patients were those who had never received ET previously or those who had primary endocrine resistance if a relapse occurred within 2 years after adjuvant endocrine therapy or if progression occurred within 6 months during the first-line ET in advanced cancer. Secondary endocrine resistance was considered when primary resistance was excluded.

### 2.3. Outcome Evaluation

The primary outcome variable was PFS, defined as the time from the onset of PAL plus ET until physician-documented disease progression or death due to any cause, whichever occurred first. The secondary outcome variables were ORR (defined as the proportion of patients with the best response (BR) of either complete response (CR) or partial response (PR) from the onset of therapy until PD), OS (defined as the time from the start of PAL plus ET until death due to any cause), patient treatment response with PAL plus ET, and the safety assessment of PAL in the overall population and the elderly population (≥65 years). Adverse events were considered according to the National Cancer Institute Common Terminology Criteria for Adverse Events (CTCAE version 5.0) [21].

### 2.4. Statistical Analyses

The demographic characteristics and adverse events of the individuals in the study were analyzed using the chi-square (χ^2^) test or rank-sum test. Kaplan–Meier estimates were used to compare PFS obtained from the log-rank test. ORRs were compared using the chi-square (χ^2^) test. The individual effects and interactions of various covariates on PFS were analyzed by Cox regression modeling. A two-tailed test with a *p*-value of <0.05 was regarded as statistically significant in all analyses. All statistical data were analyzed using Stata/SE 13.1 software (StataCorp LLC, College Station, TX, USA) to determine the baseline characteristics, treatment effectiveness, and safety.

## 3. Results

### 3.1. Basic Characteristics of Patients

A total of 397 patients from eight clinical sites were treated with palbociclib plus ET, and data were collected from electronic medical records for analysis. The baseline demographic characteristics and disease features of the patients are summarized in Table 1 and Appendix A. The median age of the patients was 55 years (range, 28–85 years), and 26.2% (n = 104) were aged ≥65 years. Most patients were postmenopausal (66.50%), and 80.60% were in good condition with an ECOG performance status of 0–1. The most common pathological type was infiltrating ductal carcinoma (194 patients, 48.87%). On the last pathological examination before therapy administration, 242 (≥20%) patients had high PR expression, 142 patients displayed low PR expression, and 7 patients had unknown PR expression. Ninety patients (22.67%) were diagnosed with de novo metastatic disease. Approximately half of the patients had visceral metastasis (49.87%), 98 patients had liver metastasis, 144 patients displayed lung metastasis, 72 patients (18.14%) had single bone metastasis, and 210 patients (52.90%) had two or more metastatic sites. Fifty-one (12.85%) patients did not receive adjuvant chemotherapy. ET was administered to 250 participants (62.97%). Seventy-two patients showed primary endocrine resistance, 231 showed secondary resistance, 81 were ET naïve, and endocrine resistance was not reported in 13 patients.

### 3.2. Treatment Patterns

Twenty-eight patients received PAL before 2019, 145 in 2019, 130 patients in 2020, and 91 patients in 2021. Among all subjects, PAL plus ET was administered as first-line (1 L) therapy for ABC in 21 (54.66%) patients and as ≥2 L therapy in 180 (45.34%) patients (Appendix A). The treatment regimens were divided into PAL plus AI (aromatase inhibitors, including letrozole, anastrozole, or exemestane), PAL plus FUL (fulvestrant), and other PAL (defined as PAL plus another hormonal therapy or PAL monotherapy). PAL plus AI was used as a 1L therapy in almost twice as many patients (n = 135) as PAL plus FUL (n = 70). For 2 L therapy, the number of patients administered with PAL plus AI (n = 91) and PAL plus FUL (n = 76) was similar. Additional treatment characteristics are presented in Appendix A.

### 3.3. Treatment Effectiveness

In patients who received palbociclib, the median PFS (mPFS) was 14.2 months for the whole population (Figure 1A), 22.3 months for the 1 L setting, and 11.1 months for the subsequent-line settings (*P*_log-rank_ < 0.0001, Figure 1B). The mPFS in ET-naïve patients was not reached, while patients with primary endocrine resistance and secondary endocrine resistance showed mPFS values of 9.13 months (95% CI, 5.57–13.7) and 13.5 months (95% CI, 12.13–17.23), respectively (*P*_log-rank_ < 0.0001, Figure 1C). When the tumor response was assessed among patients with measurable disease, the ORR and CBR with PAL plus ET were 28.97% and 66.25%, respectively (Table 2).

The relationship between baseline characteristics and treatment effectiveness was evaluated. Patients who had their last pathological evaluation less than 1 year before the study entry had a longer mPFS than those who had the evaluation over a year before (22.53 months vs. 12.13 months, *P*_log-rank_ < 0.0001). Patients with low PR expression (<20%) and high Ki-67 expression (≥30%) showed significantly poorer survival (*P*_log-rank_ = 0.0065 and *P*_log-rank_ = 0.0358), and those without liver metastasis or with fewer than two metastatic sites displayed improved prognosis (*P*_log-rank_ < 0.0001 and *P*_log-rank_ = 0.0002). As expected, patients who were chemotherapy naïve (*P*_log-rank_ < 0.0001) for metastatic or recurrent cancer showed a significantly superior prognosis. No significant differences in PFS were observed in terms of age (<65 vs. ≥65 years, *P*_log-rank_ = 0.4173), BMI (<24 vs. ≥24, *P*_log-rank_ = 0.9554), chronic disease status (*P*_log-rank_ = 0.2832), menopausal status at study entry (premenopausal vs. postmenopausal, *P*_log-rank_ = 0.4318), or pathological type (invasive lobular carcinoma vs. invasive ductal carcinoma, *P*_log-rank_ = 0.7176).

The treatment options were then discussed. Patients who received AI as prior ET presented significantly worse effectiveness than those who were not treated with AI (11.73 months vs. 22.83 months, *P*_log-rank_ < 0.0001). The PAL plus AI group had a better outcome compared to the PAL plus FUL group (20.43 months vs. 10.77 months, *P*_log-rank_ < 0.0001, Figure 1D). The outcomes of the univariate analysis are presented in Table 3 and Appendix A.

### 3.4. Subgroup Analysis

To explore the contribution of different sensitivities to ET, the adjusted PFS was further analyzed in the subgroup of patients with primary and secondary treatment resistance. In the primary resistance subgroup, mPFS was 8.1 months for the 1 L setting and 10.3 months for the ≥2 L setting. Patients who received adjuvant radiotherapy had a median PFS of 7.03 months vs. 15.17 months for those who did not (*P*_log-rank_ = 0.0130). Upon univariate analysis, patients without liver metastasis (14.47 months vs. 4.8 months, *P*_log-rank_ < 0.0001), without visceral metastases (12.40 months vs. 5.47 months, *P*_log-rank_ = 0.0388), or with fewer than two metastatic sites (11.2 months vs. 5.3 months, *P*_log-rank_ = 0.0063) generally displayed a longer mPFS (Table 3). Similar results were observed in the patients with secondary endocrine resistance (Table 3). Patients with secondary endocrine resistance had an mPFS of 17.23 months vs. 7.97 months in the non-liver metastasis vs. liver metastatic arm, respectively (*P*_log-rank_ < 0.0001), and 21.83 months vs. 12.73 months in the non-visceral disease vs. visceral diseases, respectively (*P*_log-rank_ = 0.0204). Additionally, patients who had their last pathological evaluation less than 1 year before the study entry (22.53 months vs. 12.73 months, *P*_log-rank_ = 0.0018), had less than 2 L chemotherapy before PAL administration (18.3 months vs. 9.17 months, *P*_log-rank_ < 0.0001), had never received AI as prior ET (22.33 months vs. 12.07 months, *P*_log-rank_ = 0.0003), had no prior ET at the metastatic stage (26.4 months vs. 11.13 months, *P*_log-rank_ < 0.0001), or had combined therapy with AI (compared with PAL plus FUL, 17.23 months vs. 11.73 months, respectively; *P*_log-rank_ = 0.0033) showed a significantly improved prognosis.

### 3.5. Multivariate Analysis

Multivariate analysis was used to investigate risk factors that could have affected the efficiency of PAL plus ET. Statistically significant baseline variables were included in the final multivariate Cox proportional hazard model. Ki-67 expression of ≥30% vs. <15%, secondary treatment resistance vs. primary treatment resistance, liver metastasis vs. non-liver metastasis, number of sites <2 vs. ≥2, first line of ET vs. subsequent lines, and combination with AI vs. FUL plus PAL were recognized as independent prognostic factors in the model (Table 4). The interaction effect was tested to investigate the effect of one variable on the value of another. In elderly patients with liver metastasis, the line of therapy with the sensitivity of ET and previous treatment with AI and the endocrine combination with previous treatment with AI showed interaction effects (Figure 2).

### 3.6. Safety

The safety profile was incomplete. The 394 patients with evaluable safety data were categorized into two groups, aged <65 years (n = 303) and aged ≥65 years (n = 91), based on the first administration of PAL. AEs in the two groups are summarized in Table 5. The most commonly reported treatment-related adverse events (TRAEs) of any grade among the whole population were hematologic manifestations, such as neutropenia (77.92%), leukopenia (74.87%), anemia (33.50%), and thrombocytopenia (29.70%). The most common grade 3/4 adverse event was neutropenia (46.45%). The safety in the elderly population (≥65 years) was consistent with that in the younger population and did not have new safety signals. No statistically significant difference was observed in the incidence of major adverse reactions between the two age groups (all *p*-values > 0.05, Appendix A). During the first treatment cycle, most patients (82.1%) received a recommended palbociclib starting dose of 125 mg/day. Of the patients who received an initial dose of 125 mg, 54 (13.92%) underwent dose adjustment, with 50 participants eventually adjusted to 100 mg and 4 to 75 mg. Thirteen (3.35%) patients with a starting dose of 125 mg discontinued treatment because of intolerable adverse effects, which were not observed in patients with starting doses of 100 mg or 75 mg (Appendix A).

## 4. Discussion

CDK 4/6i plus ET is currently the preferred treatment for metastatic breast cancer (MBC) with HR+/HER2− as previous randomized controlled trials (RCTs) confirmed its efficacy by obtaining better PFS and OS. However, these RCTs included relatively simple demographic characteristics, which limit the generalizability of the outcomes to the entire population. To thoroughly assess the effectiveness and safety of palbociclib among patients from diverse regions and income brackets and to accurately reflect real-world treatment trends in China, the largest multi-center study to date was carried out in a Chinese patient population. This study examined the actual pattern of palbociclib treatment and provides valuable perspectives on the use of palbociclib in China and can help to improve treatment strategies for patients with cancer.

Based on the available clinical data as of October 2022, the mPFS was 14.2 months across the entire population, whereas the mOS was not achieved. The ORR and CBR with PAL plus ET were 28.97% and 66.25%, respectively. The median age of our population was 55 years, which is younger than that in previous phase III RCTs [5,22]. This may be due to the earlier onset of the disease in China [23]. Nevertheless, the study still had about a quarter of its population in the elderly group (>65 years old), which allowed us to examine potential age-related risk factors. Our findings suggest that age alone does not affect the effectiveness of treatment, but, for patients with liver metastasis, the elderly population is at a significantly higher risk of progression than the general population (HR 0.76 (95% CI, 0.06–10.48) vs. 18.89 (4.43–82.25); *p*-value for interaction was 0.023). Patients with poor ECOG performance status (≥2) or diagnosed with chronic metabolic diseases were also included in this study analysis, and neither of these showed an independent effect on treatment efficacy, which may disclose the lower side effects of PAL.

Regarding PR, 60% of the population had PR expression greater than 20% and showed a superior mPFS compared to the lower expression group, which was not observed in the multivariate analysis. The statistical difference in the entire population may be attributed to the population composition and other confounding factors. In contrast, the outcomes of the three Ki-67 index grades showed significant biological differences. After ruling out possible confounding factors, the risk of progression was 1.73 times higher in the high-expression group than in the low-expression group, and the medium-expression group did not differ from the other two groups, which is consistent with the outcomes of previous pathological studies [24]. Although the initial diagnosis stage appeared to have an impact on PFS, given the statistically significant difference in the distribution of treatment lines between recurrent and de novo metastatic breast cancers (Appendix A), this may have been a result of the patients’ treatment choices.

The treatment effectiveness could have been influenced by tumor burdens, and fewer sites of metastasis were associated with longer survival time (22.53 months (95% CI, 14.47–28.43) vs. 12.07 months (10.13–13.70); HR 1.52 (1.05–2.20); *p* = 0.028). However, liver metastasis individually drove a shorter mPFS compared to those without (21.27 months (95% CI, 14.23–21.83) vs. 7.97 months (4.97–10.73); HR 2.22 (1.41–3.48); *p* < 0.0001). Yu et al. [25] found that liver metastasis activates CD8+ T cells in systemic circulation. Within the liver, CD8+ T cells undergo apoptosis and create a systemic immune desert by reducing peripheral T cell numbers and diminishing tumor T cell diversity and function. Nevertheless, CDK 4/6i can enhance autoimmunity by promoting the infiltration of CD8+ T cells and suppressing immunosuppressive CD4FOXP3 regulatory T cell (Treg) proliferation to decrease the Treg:CD8 T cell ratio. Animal models have demonstrated that tumor regression mediated by CDK 4/6 inhibition is, at least in part, dependent on the presence of CTLs [26]. These results may explain why the outcomes were worse in patients with liver metastasis.

The line of therapy and sensitivity of ET could have independently affected treatment effectiveness, which is consistent with previous clinical trials [4,11]. As the line of therapy increased, the gap between secondary resistance and primary resistance decreased; therefore, we suggest that patients with secondary resistance should take CDK 4/6i earlier to maximize effectiveness.

For combinatory therapy, this study attempted to determine the best combination of CDK 4/6 inhibitors, AI or FUL. Although AI use had a greater impact on progression risk for PAL combined with AI vs. FUL (HR 2.04 (95% CI, 1.03–4.09) vs. 0.9 (0.51–1.59); *p*-value for interaction was 0.023), both univariate and multivariate analyses indicated that AI had a longer median PFS than FUL (20.43 months (95% CI, 14.23–25.6) vs. 10.77 months (8.23–12.83); HR 1.40 (1.01–1.99); P_cox_ = 0.047). Among previous studies, only MONALEESA-3 showed that ribociclib combined with FUL had a longer median PFS than the combination with AI in first-line treatment [27]. Although the PARSIFAL phase II study did not show a difference between the two, their combination with AI had a numerical advantage [28]. A real-world study, POLARIS, also showed a numerical advantage in combination with AI as first-line treatment [29]. These findings overturn previous anecdotal evidence and show that combining AI with PAL might achieve better treatment effectiveness. However, a prescription bias may have occurred in which doctors may have tended to administer FUL to patients with more severe conditions who might have already had higher progression rates. Consequently, a rigorous head-to-head RCT is needed to substantiate the outcomes of better combination therapies.

This study also demonstrated the safety of PAL. One of the most common adverse events observed in all grades and grade 3/4 was neutropenia, with incidence rates of 77.92% and 46.45%, respectively. Compared with the PALOMA-4 study, our real-world study showed a lower incidence of hematological toxicity [7]. This finding suggests that PAL is safer than previously thought. The safety of PAL was consistent across all age groups, including in those aged ≥65 years. No new safety signals were identified, and no statistically significant difference was observed in the incidence of major adverse reactions between the older and younger age groups (all *p*-values > 0.05). In summary, the toxicity observed in this study was mild and easy to manage.

Our study certainly had potential limitations, which might have been mainly due to the nature of the research and the data itself. Without randomized grouping, the differences between the groups may have persisted, leading to bias. While we used complex statistical methods to eliminate known confounding factors, unknown confounders could not be entirely removed. The completeness and accuracy of the data may pose another issue. Therefore, our study aims to provide a reference from real-world clinical data for further research.

## 5. Conclusions

In summary, this work described the disease features of the patients as well as consolidated palbociclib efficacy and safety in Chinese patients. Palbociclib’s therapeutic benefits for breast cancer may be hindered by higher Ki-67 expression, primary resistance to ETs, liver metastases, multiple metastatic sites, later line of therapy, and the use of fulvestrant instead of aromatase inhibitors. Further validation in randomized controlled studies is warranted.

## Figures and Tables

**Figure 1 cancers-15-04360-f001:**
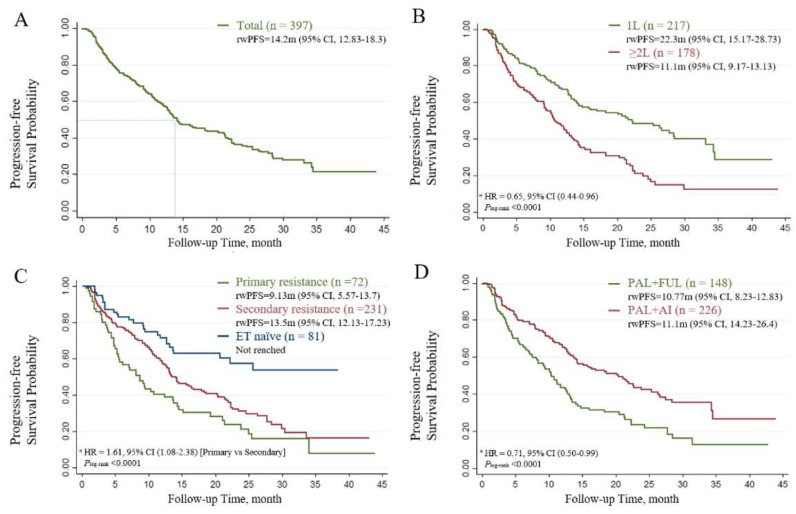
The Kaplan–Meier curves of progression-free survival in patients treated with PAL. (**A**) Survival outcomes of the entire population. (**B**) Patients receiving different lines of therapy (LoT). (**C**) Patients with different sensitivity statuses. (**D**) Patients receiving different combinations of therapies. HR, hazard ratio *; rwPFS, real-world progression-free survival. * Hazard ratios were estimated using multivariate Cox regression models adjusted for baseline demographic and clinical variables.

**Figure 2 cancers-15-04360-f002:**
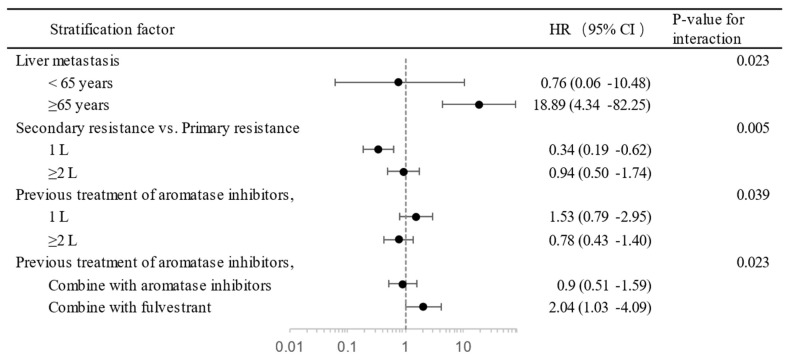
The interaction effect between independent factors of PAL treatment. Circle, Hazard Ratio (HR); dash line, 95% Confident Index.

**Table 1 cancers-15-04360-t001:** Patient baseline demographic and disease characteristics.

Characteristic	Patients (n = 397)
Age	
Median (range)	55 (28–85)
Age, n (%)	
<65 years	293 (73.80)
≥65 years	104 (26.20)
Menopausal status at study entry, n (%)	
Premenopausal	91 (22.92)
Perimenopausal	43 (10.83)
Postmenopausal	263 (66.25)
ECOG performance status, n (%)	
0–1	320 (80.60)
≥2	77 (19.40)
Disposition of diagnosis, n (%)	
De novo, newly diagnosed stage IV	90 (22.67)
Recurrent from earlier stages, stages I–III	307 (77.33)
Pathology	
Invasive ductal carcinoma (IDC)	194 (48.87)
Invasive lobular carcinoma (ILC)	51 (12.85)
Other	152 (38.29)
Expression of progesterone receptor (PR)	
<20%	148 (37.28)
≥20%	242 (60.96)
Unknown	7 (1.76)
Bone involvement, n (%)	
Bone + other metastases	144 (36.27)
Bone only	72 (18.14)
Visceral disease ^†^, n (%)	
No	198 (49.87)
Yes	199 (50.13)
Liver metastases	98 (24.69)
Lung metastases	144 (36.27)
Number of sites of metastasis	
Median (range)	2 (1.73–1.96)
Distribution, n (%)	
<2	187 (47.10)
≥2	210 (52.90)
Prior adjuvant chemotherapy, n (%)	
Yes	252 (63.48)
No	51 (12.85)
Not reported	4 (1.01)
Prior adjuvant ET, n (%)	
Yes	253 (63.73)
No	50 (12.59)
Not reported	4 (1.01)
Sensitivity to ET *, n (%)	
Primary resistance	72 (18.14)
Secondary resistance	231 (58.19)
ET naïve	81 (20.40)
Not reported	13 (3.27)
Line of ET before CDK 4/6i treatment	
1L	217 (54.66)
≥2L	178 (44.84)
Unknown	2 (0.50)

1L, first line; 2L, second line; ET, endocrine therapy. ^†^ Metastases of brain, liver, and/or lung/pleura. * The definition of sensitivity to ET is consistent with ABC6. Primary endocrine resistance exists if a relapse occurs within 2 years after adjuvant endocrine therapy (ET) or if progression occurs within 6 months during first-line ET in advanced cancer. If primary resistance is excluded, secondary endocrine resistance can be assumed.

**Table 2 cancers-15-04360-t002:** Effectiveness of PAL.

Effectiveness	n = 397
Complete response (CR)	4 (1.01%)
Partial response (PR)	111 (27.96%)
Stable disease (SD)	198 (49.87%)
Progressive disease (PD)	65 (16.37%)
Invaluable (NE)	5 (1.26%)
Not assessed (lost to follow up)	14(3.53%)
Overall response rate (ORR)	28.97%
Clinical benefit rate (CBR)	66.25%

PAL, palbociclib.

**Table 3 cancers-15-04360-t003:** Univariate analysis for independent factors of PAL.

Characteristics	All (n = 397)		Primary Resistance (n = 72)			Secondary Resistance (n = 231)		
	mPFS (mo)	*p*-Values	n (%)	mPFS (mo)	*p*-Values	n (%)	mPFS (mo)	*p*-Values
Age		0.4173			0.6556			0.3791
<65 years	14.23		55 (76.39%)	7.13		173 (74.89%)	13.5	
≥65 years	14.17		17 (23.61%)	11.2		58 (25.11%)	14.17	
The time from the last pathology to the entry		<0.0001			0.8745			0.0018
<1 year	22.53		37 (51.39%)	8.73		97 (41.99%)	22.53	
≥1 year	12.13		35 (48.61%)	9.5		134 (58.01%)	12.73	
Expression of progesterone receptor (PR)		0.0065			0.2188			0.1205
<20%	12.13		34 (47.22%)	7.03		95 (41.13%)	12.13	
≥20%	18.33		38 (52.78%)	9.13		133 (57.58%)	16.4	
Ki-67 index		0.0358			0.9236			0.0745
Low (<15%)	25.4		11 (15.28%)	7.03		42 (18.18%)	25.6	
Medium (15–30%)	13.27		29 (40.28%)	9.5		77 (33.33%)	13.13	
High (>30%)	13.7		31 (43.06%)	8.73		87 (37.66%)	13.2	
Disposition of diagnosis		0.0115			0.0419			0.1375
De novo, newly diagnosed stage IV	24.07		6 (8.33%)	21.33		30 (12.99%)	11.23	
Recurrent from earlier stage, stages I–III	13.43		66 (91.67%)	8.17		201 (87.01%)	14.23	
Prior adjuvant radiotherapy, n (%)		0.0076			0.013			0.8937
Yes	12.4		43 (59.72%)	7.03		104 (45.02%)	14.23	
No	18.3		28 (38.89%)	15.17		120 (51.95%)	13.27	
Sensitivity to ETs		<0.0001						
Primary resistance	9.13							
Secondary resistance	13.5							
ET naïve	Not reached							
Site of metastases								
Liver metastases		<0.0001			<0.0001			0.0001
Yes	7.97		22 (30.56%)	4.8		63 (27.27%)	7.97	
No	21.27		50 (69.44%)	14.47		168 (72.73%)	17.23	
Bone-only metastasis		0.101			0.8885			0.0755
Yes	13.5		19 (26.39%)	9.13		43 (18.61%)	28.43	
No	21.33		53 (73.61%)	9.5		188 (81.39%)	13.2	
Visceral disease		0.0172			0.0388			0.0204
Yes	13.13		33 (45.83%)	5.47		111 (48.05%)	12.73	
No	20.43		39 (54.17%)	12.4		120 (51.95%)	21.83	
Number of sites of metastasis		0.0002			0.0063			0.0001
<2	22.53		45 (62.50%)	11.2		97 (41.99%)	28.43	
≥2	12.07		27 (37.50%)	5.3		134 (58.01%)	11.23	
Line of chemotherapy before CDK 4/6i treatment		<0.0001			0.1582			<0.0001
Untreated	21.27		51 (70.83%)	9.13		161 (69.70%)	18.3	
1L	17.23							
≥2L	9.17		21 (29.17%)	6.13		68 (29.44%)	9.17	
Re-radiotherapy before CDK 4/6i treatment		0.007			0.8194			0.0462
Yes	12.03		8 (11.11%)	10.3		38 (16.45%)	12.53	
No	15.17		59 (81.94%)	8.73		179 (77.49%)	14.5	
Line of ET before CDK 4/6i treatment		<0.0001			0.1789			<0.0001
1L	22.33		30 (41.67%)	8.1		94 (40.69%)	26.4	
≥2L	10.73		42 (58.33%)	10.3		137 (59.31%)	11.13	
Previous treatment of AI		<0.0001			0.8546			0.0003
Yes	11.73		21 (29.17%)	9.13		89 (38.53%)	12.07	
No	22.83		51 (70.83%)	8.17		141 (61.04%)	22.33	
Combination therapy		<0.0001			0.0555			0.0033
Fulvestrant	10.77		35 (48.61%)	8.17		91 (39.39%)	11.73	
Aromatase inhibitors	20.43		31 (43.06%)	14.47		129 (55.84%)	17.23	

**Table 4 cancers-15-04360-t004:** Multivariate Cox regression analysis for independent factors of PAL treatment.

Characteristic	HR	95% CI	*p*-Value
Ki-67 status			
≥30% vs. <15%	1.73	1.11–2.70	0.015
Sensitivity to ET			
Primary resistance vs. secondary resistance	1.61	1.08–2.38	0.018
Liver metastases			
Yes vs. no	2.22	1.41–3.48	0.001
Number of sites of metastasis			
≥2 vs. <2	0.66	0.45–0.95	0.028
LOT (line of therapy) of ET			
1 L vs. ≥2 L	0.65	0.44–0.96	0.031
Endocrine partner			
Aromatase inhibitors vs. fulvestrant	0.71	0.50–0.99	0.047

**Table 5 cancers-15-04360-t005:** Treatment-related adverse events.

	All (n = 394)		<65 Years Old (n = 303)		≥65 Years Old (n = 91)	
	All Grades, n (%)	Grade 3/4, n (%)	All Grades, n (%)	Grade 3/4, n (%)	All Grades, n (%)	Grade 3/4, n (%)
Neutrophil count decreased	307 (77.92%)	183 (46.10%)	235 (77.56%)	152 (50.17%)	72 (79.12%)	31 (34.07%)
White blood cell decreased	295 (74.87%)	114 (28.93%)	229 (75.58%)	94 (31.02%)	66 (72.53%)	20 (21.98%)
Anemia	132 (33.50%)	16 (4.06%)	95 (31.35%)	13 (4.29%)	37 (40.66%)	3 (3.30%)
Platelet count decreased	117 (29.70%)	24 (6.09%)	89 (29.37%)	19 (6.27%)	28 (30.77%)	5 (5.49%)
Aspartate aminotransferase increased	17 (4.31%)	3 (0.76%)	14 (4.62%)	2 (0.66%)	3 (3.30%)	1 (1.10%)
Malaise	16 (4.06%)	-	11 (3.63%)	-	5 (5.49%)	-
Alanine aminotransferase increased	14 (3.55%)	2 (0.51%)	11 (3.63%)	1 (0.33%)	3 (3.30%)	1 (1.10%)
Infections and infestations	9 (2.28%)	1 (0.25%)	5 (1.65%)	-	4 (4.40%)	1 (1.10%)
GGT increased	8 (2.03%)	1 (0.25%)	6 (1.98%)	-	2 (2.20%)	1 (1.10%)
Diarrhea	6 (1.52%)	1 (0.25%)	4 (1.32%)	-	2 (2.20%)	1 (1.10%)
Rash maculo-papular	6 (1.52%)	-	6 (1.98%)	-	-	-
Vomiting	5 (1.27%)	-	1 (0.33%)	-	4 (4.40%)	-
Alkaline phosphatase increased	4 (1.02%)	-	3 (0.99%)	-	1 (1.10%)	-
Mucositis oral	4 (1.02%)	-	3 (0.99%)	-	1 (1.10%)	-
Fatigue	4 (1.02%)	-	3 (0.99%)	-	1 (1.10%)	-
Cholesterol high	4 (1.02%)	-	2 (0.66%)	-	2 (2.20%)	-
Dizziness	3 (0.76%)	-	3 (0.99%)	-	-	-
Cough	3 (0.76%)	-	3 (0.99%)	-	-	-
Pruritus	2 (0.51%)	-	1 (0.33%)	-	1 (1.10%)	-
Nausea	2 (0.51%)	-	1 (0.33%)	-	1 (1.10%)	-
Blood lactate dehydrogenase increased	2 (0.51%)	-	2 (0.66%)	-	-	-
Hypertriglyceridemia	2 (0.51%)	-	1 (0.33%)	-	1 (1.10%)	-
Hyperuricemia	2 (0.51%)	-	1 (0.33%)	-	1 (1.10%)	-
Lymphocyte count decreased	2 (0.51%)	-	2 (0.66%)	-	-	-
Constipation	2 (0.51%)	-	1 (0.33%)	-	1 (1.10%)	-
Hypoalbuminemia	1 (0.25%)	-	-	-	1 (1.10%)	-
Palmar–plantar erythrodysesthesia syndrome	1 (0.25%)	-	1 (0.33%)	-	-	-
Hyperhidrosis	1 (0.25%)	-	1 (0.33%)	-	-	-

## Data Availability

Data supporting the findings of this study are available from Y. Yin upon reasonable request.

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
