# Peer review of "Effectiveness and Safety of Palbociclib Plus Endocrine Therapy in Patients with Advanced Breast Cancer: A Multi-Center Study in China"

_cancers, 2023, doi:10.3390/cancers15174360_

Round 1

Reviewer 1 Report

This is an excellent review of the role of palbociclib in Chinese patients with metastatic breast cancer. however, the details may not be of particular interest to the rest-of-the-world. I still recommend its publication, as researchers may derive new ideas and basis for new research; it may be in an abridged form and more details in appendices.

Palbociclib is well known by the clinician and there is nothing new in the paper, except that it solidifies palbociclib efficacy and safety in Chinese patients.

Author Response

Thank you for your valuable feedback. We have revised the paper with the help of a native English speaker to improve its readability. Palbociclib is well-known by clinicians, and this article has certain limitations in terms of innovation. We will explore further directions based on current research. The co-authors and I sincerely appreciate the time and effort you spent reviewing the manuscript.

Reviewer 2 Report

1. Abstract line 23 and introduction line 75-you use term latent factors. Please define what that term means-I am not clear

2.Introduction Line 44-improved prognosis- compared to what? Maybe you want to say it has a favorable prognosis compared to other subtypes of breast cancer?

3. Introduction-the sentence on lines 47-49 does not make sense-needs to be rewritten.

4.Introduction lines 54-55 improved survival compared with what? I think you mean to say when added to endocrine therapy

5. Introduction lines 59-60-please define the treatment arms for PALOMA 3

6. Table 1 - you list the following. What is in the Else section? What other histologic types are you referring to?

Pathology
Invasive ductal carcinoma (IDC) 194 (48.87)
Invasive lobular carcinoma (ILC) 51 (12.85)
Else 152 (38.29)

7.Patients- line 99-you say "not amenable to treatment"-that statement is not clear, and the patient population is not clear. Did you mean to say they have incurable disease? And to clarify further-did you include locally advanced disease or only stage IV?

8. Patients line 157 says the following:

"most common pathological type was infiltrating ductal carcinoma (195 patients, 49.12%)"

If this accounts for only 49% of patients, which is less than half, and is the most common type, what are the other types?  And this figure does not quite match  table 1 which has 194 patients with ductal.

9.Patients line 159 says the following. I think you meant to say they had de novo metastatic disease?

"Ninety patients (22.67%) were diagnosed with stage â…£ disease"

10. Results line 200- you discuss last pathologic evaluation. I do not understand the significance of that. Many patients will have biopsies at disease progression. Please clarify what that means and why it is relevant.

11. Results line 213- please clarify what this sentence means-worse compared to what? What is being compared?

"Patients who had received AI as prior ET presented significantly worse effectiveness "

English is generally good-suggestions are made in my comments

Author Response

Thank you for taking the time and effort to review the manuscript. I have uploaded a point-by-point response to your comments as a Word file. Please check the attachment. Additionally, I have provided responses to the reviewers' comments below:

Point 1. Abstract line 23 and introduction line 75-you use term latent factors. Please define what that term means-I am not clear
Response 1: latent means (state)existing but not yet developed or manifest; hidden or concealed. This term I used in line 23 attempted to express that some factors are not prominent when observed directly and can only be inferred indirectly through a mathematical model. I have already changed the “latent factors” into “latent variables”, hope that expresses more clearly what I am trying to say. The phrase in Line 75 already changed to factors
Point 2. Introduction Line 44-improved prognosis- compared to what? Maybe you want to say it has a favorable prognosis compared to other subtypes of breast cancer?
Response 2: Yes, thank you for pointing that out, already changed this sentence to “Hormone receptor (HR) positive and human epidermal growth factor receptor-2 (HER2) negative (HR+/HER2-) breast cancer is one of the most diagnosed cancers in women and generally has a favorable prognosis compared to other subtypes of breast cancer.”
Point 3. Introduction-the sentence on lines 47-49 does not make sense-needs to be rewritten.
Response 3: Thank you, already deleted.
Point 4. Introduction lines 54-55 improved survival compared with what? I think you mean to say when added to endocrine therapy
Response 4: Already changed to “Multiple clinical trials have found statistically significant improvements in progres-sion-free survival (PFS) and overall survival (OS) when added CDK 4/6 inhibitors to endocrine therapy.” 
Point 5. Introduction lines 59-60-please define the treatment arms for PALOMA 3
Response 5: The treatment arm is palbociclib (PAL) and fulvestrant (FUL), already changed this sentence to “The PALOMA-3 trial proved that the combination of PAL and fulvestrant (FUL) improved PFS compared to placebo plus FUL.” 
Point 6. Table 1 - you list the following. What is in the Else section? What other histologic types are you referring to?
Pathology
Invasive ductal carcinoma (IDC) 194 (48.87)
Invasive lobular carcinoma (ILC) 51 (12.85)
Else 152 (38.29)
Response 6: “Else” refers to mucinous carcinoma, medullary carcinoma, metaplastic carcinoma, adenoid cystic carcinoma, etc. Also, in some cases, the metastasis site’s biopsy pathology is only reported as the invasive carcinoma, due to poor differentiation, was included in “else”.
Point 7. Patients- line 99-you say "not amenable to treatment"-that statement is not clear, and the patient population is not clear. Did you mean to say they have incurable disease? And to clarify further-did you include locally advanced disease or only stage IV?
Response 7: This study included locally advanced disease, to be more clear, this sentence was changed to “Patients who had a histopathologically or imaging-confirmed diagnosis of invasive locally advanced or metastasis breast cancer”
Point 8. Patients line 157 says the following:
"most common pathological type was infiltrating ductal carcinoma (195 patients, 49.12%)"
If this accounts for only 49% of patients, which is less than half, and is the most common type, what are the other types?  And this figure does not quite match table 1 which has 194 patients with ductal.
Response 8: Sorry, there is a typo, we have 194 patients with ductal, already corrected that. the “else” means pathological types other than invasive ductal and lobular carcinoma, and those with poor differentiation can not be defined as certain types of carcinoma. 
Point 9.Patients line 159 says the following. I think you meant to say they had de novo metastatic disease?
"Ninety patients (22.67%) were diagnosed with stage â…£ disease"
Response 9: Yes, already corrected as "Ninety patients (22.67%) were diagnosed with de novo metastatic disease"

Point 10. Results line 200- you discuss last pathologic evaluation. I do not understand the significance of that. Many patients will have biopsies at disease progression. Please clarify what that means and why it is relevant.
Response 10: Not every time patients take biopsies when the disease progression and the pathology can be changed as the treatment goes on, so some patients may receive PAL directly without another pathological biopsy after long-term endocrine therapy. Here we want to know if the pathology is not confirmed for a long time, whether it will have a possible impact on the effectiveness.
Point 11. Results line 213- please clarify what this sentence means-worse compared to what? What is being compared?
"Patients who had received AI as prior ET presented significantly worse effectiveness "
Response 11: already corrected with "Patients who had received AI as prior ET presented significantly worse effectiveness than those who had not been treated with AI."
